# Maternal Uniparental Disomy of Chromosome 20 (UPD(20)mat) as Differential Diagnosis of Silver Russell Syndrome: Identification of Three New Cases

**DOI:** 10.3390/genes12040588

**Published:** 2021-04-17

**Authors:** Pierpaola Tannorella, Daniele Minervino, Sara Guzzetti, Alessandro Vimercati, Luciano Calzari, Giuseppa Patti, Mohamad Maghnie, Anna Elsa Maria Allegri, Donatella Milani, Giulietta Scuvera, Milena Mariani, Piergiorgio Modena, Angelo Selicorni, Lidia Larizza, Silvia Russo

**Affiliations:** 1Research Laboratory of Medical Cytogenetics and Molecular Genetics, IRCCS Istituto Auxologico Italiano, 20095 Milan, Italy; p.tannorella@auxologico.it (P.T.); daniele.minervino@outlook.com (D.M.); guzzetti.sa@gmail.com (S.G.); alevime19@gmail.com (A.V.); l.calzari@auxologico.it (L.C.); l.larizza@auxologico.it (L.L.); 2Department of Pediatrics, IRCCS Istituto Giannina Gaslini, 16147 Genova, Italy; giusypattis@gmail.com (G.P.); mohamadmaghnie@gaslini.org (M.M.); annaallegri@gaslini.org (A.E.M.A.); 3Department of Neuroscience, Rehabilitation, Ophthalmology, Genetics, Maternal and Child Health (DINOGMI), University of Genova, 16132 Genova, Italy; 4Pediatric Highly Intensive Care Unit, Fondazione IRCCS Ca’ Granda Ospedale Maggiore Policlinico, 20122 Milan, Italy; donatella.milani@policlinico.mi.it (D.M.); giulietta.scuvera@policlinico.mi.it (G.S.); 5Medical Genetics Unit, Woman-Child-Newborn Department, Fondazione IRCCS Ca’ Granda-Ospedale Maggiore Policlinico, via Francesco Sforza 28, 20122 Milan, Italy; 6UOC Pediatria, ASST Lariana, 22100 Como, Italy; milena.mariani@asst-lariana.it (M.M.); angelo.selicorni@asst-lariana.it (A.S.); 7SOS-ID Laboratorio di Genetica, ASST Lariana, 22100 Como, Italy; piergiorgio.modena@asst-lariana.it

**Keywords:** Silver Russell, growth disorder, epigenetic deregulation, rare mechanisms, UPD(20)mat, Mulchandani–Bhoj–Conlin syndrome, *GNAS* DMR, diagnostic flowchart

## Abstract

Silver Russell Syndrome (SRS, MIM #180860) is a rare growth retardation disorder in which clinical diagnosis is based on six features: pre- and postnatal growth failure, relative macrocephaly, prominent forehead, body asymmetry, and feeding difficulties (Netchine–Harbison clinical scoring system (NH-CSS)). The molecular mechanisms consist in (epi)genetic deregulations at multiple loci: the loss of methylation (LOM) at the paternal *H19/IGF2*:IG-DMR (chr11p15.5) (50%) and the maternal uniparental disomy of chromosome 7 (UPD(7)mat) (10%) are the most frequent causes. Thus far, about 40% of SRS remains undiagnosed, pointing to the need to define the rare mechanisms in such a consistent fraction of unsolved patients. Within a cohort of 176 SRS with an NH-CSS ≥ 3, a molecular diagnosis was disclosed in about 45%. Among the remaining patients, we identified in 3 probands (1.7%) with UPD(20)mat (Mulchandani–Bhoj–Conlin syndrome, OMIM #617352), a molecular mechanism deregulating the *GNAS* locus and described in 21 cases, characterized by severe feeding difficulties associated with failure to thrive, preterm birth, and intrauterine/postnatal growth retardation. Our patients share prominent forehead, feeding difficulties, postnatal growth delay, and advanced maternal age. Their clinical assessment and molecular diagnostic flowchart contribute to better define the characteristics of this rare imprinting disorder and to rank UPD(20)mat as the fourth most common pathogenic molecular defect causative of SRS.

## 1. Introduction

Silver Russell Syndrome (SRS, MIM #180860) is a rare (1:30,000–1:100,000) growth retardation condition characterized by a wide spectrum of signs and symptoms. Clinical diagnosis of SRS is currently based on the Netchine–Harbison clinical scoring system (NH-CSS), a combination of six recurrent characteristic features: pre- and postnatal growth failure, relative macrocephaly, prominent forehead, body asymmetry, and feeding difficulties [1]. Patients with at least four (out of six) criteria are defined as “clinical SRS”, even if no molecular anomaly is detected: however, molecular testing is recommended in patients with ≥3/6 criteria [1,2].

The etiology of SRS involves genetic and epigenetic mechanisms acting in various chromosomal regions subject to imprinting [1,3], with a high recurrence among the genetic alterations of uniparental disomy (UPD). Both maternal and paternal UPDs have been described for all human chromosomes except 19 and Y; they occur when both homolog chromosomes are inherited by a single parent [4]. When UPD encompasses an imprinted locus, the overall balance of methylation between parental alleles and consequent gene expression is deregulated.

The most frequent alterations are loss of methylation (LOM) of the paternal allele at *H19/IGF2*:IG-DMR (chr11p15.5), accounting for over 50% of patients, followed by maternal uniparental disomy of chromosome 7 (UPD(7)mat) and epimutations deregulating *GRB10*:alt-TSS-DMR, *PEG10*:TSS-DMR, and *MEST:*alt-TSS-DMR [5,6], which are detected in about 10% of cases.

A minor contribution to SRS-like presentation and differential diagnoses is represented by rare (epi)genetic imbalances driven by diverse chromosomal rearrangements [7]; chromosome 14q32 mat(UPD) and epimutation at the *DLK1/MEG3* region (Temple syndrome, MIM # 616222); rare pathogenic variants within the *CDKN1C*, UPD(16)mat, and UPD(20)mat; and defects of the *HMGA2*–*PLAG1*–*IGF2* pathway [8], as indicated in the SRS consensus flowchart [1].

UPD(20)mat (Mulchandani–Bhoj–Conlin syndrome (MBCS), OMIM #617352) is a poorly characterized condition: only 18 non-mosaic and 3 mosaic cases, including a description of their clinical phenotypes [9,10,11,12,13,14,15,16,17] have been reported. The patients shared severe feeding difficulties associated with failure to thrive and intrauterine/postnatal growth retardation.

UPD(20)mat causes the loss of the expressed paternal allele of the *GNAS* locus at 20q13.22. The organization and expression of maternal and paternal alleles of the *GNAS* complex locus are illustrated in Figure 1. Due to four alternative promoters and 5’ exons, multiple transcripts including *Gsα*, encoding the α-subunit of the stimulatory guanine nucleotide-binding protein (G protein), and the extra-large forms of Gsα (*XLαs*), *A/B*, and *NESP55* are produced. Gsα is a signaling protein component of the cyclic adenosine monophosphate (cAMP) pathway that is essential for the actions of parathyroid hormone (PTH) and many other hormones. Paternal UPD(20) leads to a reduced expression of Gsα, resulting in PTH resistance and pseudohypoparathyroidism (PHP) type 1b [18,19,20]. Except for *Gsα*, subject to tissue-specific imprinting, the remaining transcripts originate from a single parental allele in all tissues: *NESP55* is expressed from the maternally derived allele, whereas *XLαs* and *A/B* are expressed from the paternal one.

Conversely, the pathophysiological mechanisms underlying the clinical manifestations of UPD(20)mat remain incompletely understood, though a few studies hinted at the possible implication of the *GNAS* locus in feeding, growth, and energy metabolism [21,22,23,24]. It has been suggested that UPD(20)mat may be associated with the loss of paternal *GNAS* gene products, including *XLas* and *A/B,* and hypersensitivity of Gsα-mediated hormone receptor [15].

The relevance of UPD(20)mat in the etiology of SRS and the priority to assign this genetic/epigenetic defect across the diagnostic flowchart for the syndrome are not yet defined, as the literature offers controversial data on its incidence among SRS probands negative to the most common alterations [13,14,15,17].

We add to this limited background the clinical and molecular reports of three new patients with UPD(20)mat who underwent molecular screening for SRS because of NH-CSS ≥ 3. This study aims to further define the common features shared by patients affected by this rare condition and the UPD(20)mat prevalence in the clinically diagnosed SRS/SRS-like disorder.

## 2. Materials and Methods

### 2.1. Population Study

A cohort of 176 patients (93 boys and 83 girls), scored as NH-CSS ≥ 3, was referred to our center for SRS genetic testing. The study was approved by the ethical committee of IRCSS Istituto Auxologico Italiano. The probands’ parents received information about the study and signed a consent form. Clinical specialists examined each patient and recorded the physical and clinical characteristics as present or absent from an extensive list of SRS/SRS-like clinical features. Clinical scores based on the NH-CSS features [1] were generated using the same clinical datasheet.

### 2.2. Molecular Analysis and Study Design

After DNA extraction from whole blood, the samples entered the diagnostic flowchart investigating first the methylation status at the imprinted chromosomal region 11p15 using methylation-specific multiplex ligation probe-dependent amplification (MS-MLPA) (ME030 BWS/SRS, MRC-Holland). In cases showing a proper biparental contribution, a second MS-MLPA assay was applied (ME032-UPD7-UPD14, MRC-Holland) to detect the methylation status at the 6q24.2, 7p12.1, 7q32.2, and 14q32.2 regions. The patients who were indicative for UPD(7)mat or UPD(14)mat were analyzed with microsatellite (Short Tandem Repeat (STR)) markers both to confirm UPD and to define its extension. Then, in order to detect uncommon molecular defects, the samples showing a normal methylation profile were tested by MS-MLPA for chromosome 20 (ME031-GNAS, MRC-Holland); STR analysis for chromosome 20 was finally applied to confirm UPD occurrence.

For patient 1, an array was performed using SurePrint G3 CGH+SNP 4×180K (Agilent Technologies) whole-genome arrays using DNA extracted from peripheral blood and sex-matched reference DNA (Agilent) following the manufacturer’s instructions. The chips were scanned on an Agilent SureScan Microarray Scanner, and CytoGenomics 4.0.3.12 software (Agilent) was used to identify DNA copy number anomalies at the probe level and regions of absence of heterozygosity (AOH). The quality parameter “Derivative of log-ratio standard deviation” scored <0.3 in all experiments. 

## 3. Results

The first step in the flowchart applied to 176 patients allowed us to detect the LoM at *H19/IGF2*:IG-DMR (31.3%) in 55 patients, 11p15.5 chromosomal rearrangements in 2 cases (maternal duplication at *H19/IGF2*:IG-DMR (0.6%) and *KCNQ1OT1*:TSS-DMR (0.6%)), and duplication of *NSD1* gene (5q35) (0.6%) in one case. The second analyses disclosed 17 cases of UPD(7)mat (9.7%) and four cases altering chromosome 14 DMR (2.2%) including two UPD(14)mat, both involving the whole chromosome, as demonstrated by the parents in proband segregation analyses, and two LoM at *MEG3*-DMR (14q32) (1.1%). Furthermore, a maternal duplication at *GRB10*-intr1 (7p12.1) (0.6%) was detected (Figure 2).

Out of the 96 patients with proper methylation status, the *GNAS* MS-MLPA (Appendix A) allowed us to identify three cases (1.7% of the initial cohort) with a LoM of *NESP55* and a hypermethylation profile at the *GNAS* locus and correct CNV number. A further microsatellite segregation analysis or SNP array confirmed for all the patients the occurrence of a non-mosaic maternal UPD 20 (Figure 3).

Table 1 sums up the clinical features of the three patients described in this study, indicating the growth parameters at birth and at the clinical evaluation within the first two years, and the presence/absence of NH-CSS criteria; additional findings such as facial dimorphisms, and skeletal and genitourinary anomalies are also reported. Severe pre-and postnatal weight and length reduction, feeding difficulties, protruding forehead, and triangular face are shared by all probands. In the last column of Table 1, data from all the UPD(20)mat patients described in the literature are provided for cross-comparison with the denominators referring to the number of patients who were studied for each specific parameter. The comparison between our results and the literature patients confirms the recurrent clinical features observed in our cases, besides protruding forehead, which was reported in 4/8 cases (50%); interestingly macrocephaly, one of the distinctive traits of SRS, is mentioned only in 4/10 cases in the literature and in 1/3 of our patients.

### 3.1. UPD(20)mat Patients

#### 3.1.1. Patient 1

Patient 1 is the third male child of non-consanguineous healthy parents; in his anamnesis, an elder sister affected by short stature related to GH deficiency is reported, but the girl was not referred to our hospital. The family history is negative for genetic diseases. Pregnancy was reported as regular. Our patient was born from a 44-year-old mother at 36 + 4 weeks of gestation after spontaneous delivery. Birth weight was 1510 Kg (−3.12 SDS for gestational age), length 43 cm (−2.27 SDS for gestational age), and head circumference 30 cm (−2.53 SDS).

Chromosomal analysis performed during his hospitalization in the neonatal intensive care unit revealed a normal 46, XY male karyotype. Initial generalized hypotonia was apparent in the neonatal period, associated with poor sucking in the first month of life, which never required a tube feeding. Psychomotor development showed then a normal acquisition of milestones (Griffiths Mental Development Scales (GMDS) at 18 months: GQ = 109), but language delay was noticed after 2 years of age (GDMS at 3 years: GQ = 67 with language delay = 47). Brain MRI was normal. No significant pediatric problems were recorded: only bilateral cryptorchidism was surgically treated at the age of 2 years.

The patient was referred to our attention at 15 months because of failure to thrive. The first clinical evaluation showed a child in generally good conditions, with a slightly asymmetric (right > left) triangular face with prominent forehead and micrognathia, blue sclerae, and mild but clinically apparent lower limbs asymmetry (right > left). His growth was regular but below the lower limits. At the first evaluation, his weight was 6.7 kg (<3rd percentile, −4.9 SDS), height 71.5 cm (>3rd percentile, −2.7 SDS), and head circumference 45 cm (3rd percentile, −1.8 SDS). He satisfied 5/6 NH-CSS criteria, with relative macrocephaly being the only absent feature. Clinical follow-up showed the persistence of low weight gain (at 2 years and 5 months, weight was 8270 Kg, <<3rd percentile and 50th percentile for 7 months of age), with a height between the 3rd and 10th percentiles and BMI of 12.65 (<3rd percentile). Due to poor growth, he underwent endocrinological evaluation and, because of growth hormone (GH) deficiency, he started GH treatment at 4 years of age, with a fairly good response. Unfortunately, the patient’s parents did not agree to publication of the child’s photos.

Tests for methylation abnormalities in the 11p15.5 region and UPD7(mat) were performed, which gave normal results. The CGH+SNP-array showed a balanced genomic male profile and a ≈21 Mb region of “absence of heterozygosity” (AOH) on chromosome 20, encompassing the pericentromeric region 20p12.1-q13.11. The ISCN nomenclature assigned is arr[GRCh37] 20p12.1q13.11(13377756_41914864)x2 hmz (Figure 3A). The comparison of SNP polymorphisms on different chromosomes was consistent with the expected parents-to-son inheritance. The occurrence of UPD20 was confirmed by MS-MLPA (M031), which includes several probes within the *GNAS/NESPAS55* region, all deregulated in our patient.

#### 3.1.2. Patient 2

Patient 2 (Figure 3 Panel B) was a boy born at 38 + 1 weeks of gestation from a 42-year-old mother. 

He had a birth weight of 2.23 kg (<3rd percentile, −2.38 SDS), length of 44 cm (<3rd percentile, −2.65 SDS), and head circumference of 33.1 cm (16th percentile, −1.01 SDS). He showed relative macrocephaly and feeding difficulties at birth. Family history was not significant. Chromosomal analysis revealed a normal 46, XY karyotype. At the age of 5.5 months, his weight was 4.15 kg (<<3rd percentile, −5 SDS), his height was 60 cm (<3rd percentile, −2.7 SDS), and his head circumference was 38.5 cm (<3rd percentile, −3.5 SDS). Furtherly, he was visited at 2 years, showing a weight of 8.9 Kg (−3.3 SDS) and a height of 80.3 cm (−1.93 SDS), and at 4.5 years, when weight (15.8 Kg, −0.7 SDS) and height (102.5 cm, −0.7 SDS) were in normal ranges. 

Facial dysmorphic features included triangular face, prominent forehead, micrognathia, epicanthus, ear helix hypoplasia, short philtrum, and thin lips. He also had fifth finger clinodactyly, without any skeletal asymmetry. He achieved an NH-CSS of 4/6. UPD20 was identified by *GNAS* MS-MLPA (M031), and heterodisomy UPD(20)mat was confirmed by STR analysis (Figure 3B).

#### 3.1.3. Patient 3

Patient 3 (Figure 3B) was born from a 40-year-old mother at 38 + 5 weeks of gestation. She had a birth weight of 2.48 kg (4th percentile, −1.79 SDS), a length of 46 cm (4th percentile, −1.7 SDS), and a head circumference of 31.5 cm (2nd percentile, −1.96 SDS). The SDS values for length and weight were close to the limits for a diagnosis of SGA (Small for Gestational Age). Her family history was noncontributory, and prenatal chromosome analysis on amniotic fluid revealed a normal 46, XX female karyotype.

At an age of 15 months, her weight was 6.52 kg (<3rd percentile, −4.9 SDS), her height was 69.9 cm (<<3rd percentile, −2.5 SDS), and her head circumference was 46 cm (50th percentile, 0.01 SDS), with a BMI SDS of −2.1. The girl displayed a small and triangular face with short palpebral fissures and prominent forehead and had vaginal synechiae. She showed muscular hypotonia/hypotrophy and no skeletal asymmetry and fulfilled 3/6 NH-CSS criteria. From the age of 9 months, she was treated for growth hormone (GH) deficiency with successful growth acceleration. During the last evaluation, at the age of 9 years and 11 months, her weight was 26.5 kg (−0.9 SDS), her height was 133.8 cm (−0.3 SDS), and her growth rate was 7.5 cm/year (+2.5 SDS). 

UPD20 was identified by *GNAS* MS-MLPA (M031). Mixed hetero- and iso-UPD(20)mat were confirmed by STR analysis (Figure 3C).

## 4. Discussion

SRS is a genetically and clinically heterogeneous condition. According to the first international consensus, its clinical diagnosis is based on the presence of six specific clinical criteria, described in the NH-CSS: when four criteria are present, the proband is diagnosed as “clinical SRS”, even if no molecular alteration is detected, whereas the threshold required to access to molecular testing is ≥3 criteria [1]. Screening of the most frequent (epi)mutation does not allow us to achieve a diagnosis in about 40% of SRS, pointing to the need to define the rare mechanisms underlying SRS/SRS-like presentation in the consistent fraction of unsolved patients.

With the aim to understand the frequency of the mechanism(s) and features of the cohorts without 11p15 and chromosome 7 anomalies, previously investigated for UPD(20)mat, we noticed relevant discrepancies among the studies: Azzi et al. [13] described a single case out of 14 (7.1%) children, all with NH-CSS ≥4; Kawashima and colleagues [15] studied two different sets of patients, 55 with SRS features and 96 small for gestational age-short stature detecting 3 UPD(20)mat in the first group (5.5%) and only 1 in the second one (1%), while in the very recent paper of Hjortshøj [17], only 2/673 cases (<<1%) were detected, but no specific details or comments on the cohort were reported.

Thanks to our cumulated large cohort of 176 patients clinically evaluated as NH-CSS ≥3, filtered from the fraction harboring the prevalent molecular anomalies (about 40%), we could identify three UPD(20)mat patients with NH-CSS values of 3, 4, and 5, respectively, accounting for 1.7% of our entire SRS cohort and 3% of the molecularly diagnosed subset. According to our data, this rare genetic mechanism is the fourth defect after LoM at *H19/IGF2*:IG-DMR and UPD(7)mat and/or chromosome 7 (epi)mutations causative of an SRS/SRS like phenotype; despite the detection rate not being as high as in some studies, the percentage of children with the SRS/SRS-like phenotype and UPD(20)maternal is not negligible, deserving to be added to the diagnostic flowchart for this syndrome to implement the diagnostic rate.

The UPD(20)mat patients herein reported displayed a phenotype highly overlapping SRS: two cases fulfilled the NH-CSS criteria to achieve the definition of “Classical SRS” and one with 3/6 criteria was suitable to enter the SRS molecular flowchart. However, the clinical findings observed in our results and the literature patients highlight a few rare but prototypic findings that may help to properly address the molecular diagnosis. Our patients displayed low birth weight, prominent forehead, feeding difficulties (and/or low BMI), and postnatal growth retardation within 24 months. Interestingly, two of them displayed growth hormone deficiency. These data confirm the pre- and postnatal growth failure and feeding difficulties previously described as main shared features by all UPD(20)mat cases [10,13,14,15,16,17]. Interestingly, only patient 1 displayed body asymmetry and only patient 2 had relative macrocephaly at birth: this latter parameter was achieved only in 10 cases among the reported ones and was present in 4 of them. Even if the number is still limited, this trait, so distinctive of SRS, is found in less than 50% of UPD(20)mat, with NH-CSS ≥ 3 SRS patients (as calculated in at least 15/21 reported patients). Furthermore, all of our patients showed small and triangular facies, 2/3 showed micrognathia, and 2/3 showed muscular hypotonia, and minor characteristics were also present with sporadic frequency in the cases described so far [10,13,14,15,16,17].

Another observation that may be helpful in recognizing UPD(20)mat patients is the advanced maternal age. Consistent with previous reports [10,13,14,15,16], our study confirms the association of maternal UPD with advanced maternal age, which is recorded > 40 years (mean maternal age = 42 years) for all three patients. The phenomenon may be explained considering that trisomy 20 mosaicism is one of the most frequent cytogenetic abnormalities observed in amniocentesis or chorionic villus sampling [25]. Approximately 90% of the offspring from these pregnancies are phenotypically normal [26]. Older mothers have an increased risk of trisomy formation, and the UPD represents a potential mechanism for aneuploidy rescue [4]. With regard to this phenomenon, one can argue that low-level residual mosaic trisomy 20 might contribute in a small amount to the phenotype of our patients, complicating the clinical implication of UPD(20)mat. However, growth retardation and failure to thrive exhibited by our patients are lacking in patients with mosaic trisomy 20 [26,27], disregarding this argument.

The phenotype of UPD(20)mat may be likely related to overexpression of *Gsα* and/or deficiency of paternally expressed *GNAS* transcripts, including *XLas* and *A/B*. *Gsα* is expressed predominantly from the maternal allele in the proximal renal tubule, thyroid, pituitary gland, and ovary and is essential for the actions of parathyroid hormone (PTH) and other hormones. Moreover, endocrinological assessment of five UPD(20)mat patients showed that this condition is associated with hypersensitivity of Gsα-mediated hormone receptor, which may gradually develop with age [15]. However, another hypothesis is supported by studies on rodents that showed how deficiency of the paternally expressed *Gnasxl* transcript impairs postnatal adaptation to feeding in newborns; indeed, mice lacking *XLas* on the paternal allele showed poor sucking and postnatal growth failure [22,23], and animals lacking paternal exon 1A (corresponding to human *A/B*) exhibited prenatal growth retardation [28].

These findings are consistent with the “parental conflict” theory of genomic imprinting. This hypothesis predicts that paternally expressed imprinted genes promote the delivery of resources to the offspring during the gestational period and that loss of these genes leads to reduced fetal and early postnatal growth [29]. The overall scenario is actually complex and cannot be disentangled without the availability of tissues other than blood for epi-transcriptomic studies of the *GNAS* locus in UPD (20) mat patients.

## 5. Conclusions

This study contributes to the definition of clinical features in UPD(20)mat patients. It is interesting to note that UPD(20)mat is one of the most common pathogenically rare mechanisms identified in our cohort of SRS/SRS-like patients with high NH-CSS values. These data support the recommendation to include the methylation analysis of chromosome 20 in the diagnostic flowchart of the SRS/SRS-like and the perspective of possible therapeutic options. Two of our UPD(20)mat patients had GH deficiency and showed significant growth acceleration after GH treatment similar to a few reported patients from the literature [14,17]. The phenotypical description of our UPD(20)mat children contributes to better define the characteristics of this rare imprinting disorder and, therefore, facilitates future diagnosis through specific updated molecular tests.

## Figures and Tables

**Figure 1 genes-12-00588-f001:**
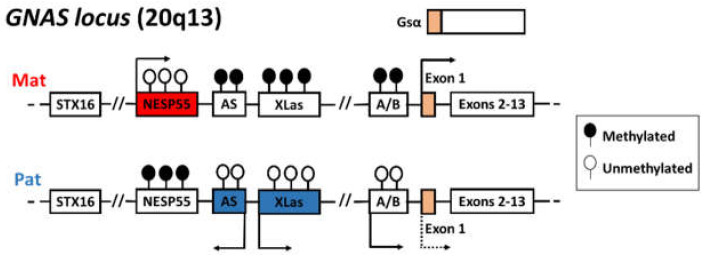
The imprinted *GNAS* locus (20q13.3) encodes multiple transcripts, including *Gsα*, *XLαs*, *NESP55,* and A/B. These isoforms have four alternative promoters and unique first exons, and they share the exons 2–13. The initiation of maternal- and paternal-specific transcripts is shown by arrows. The first exon of the *Gsα* gene is biallelically expressed in most tissues, but there is a preferable expression of the maternal allele in some specific tissues. Filled lollipops represent methylated DMRs and empty lollipops represent the unmethylated DMRs.

**Figure 2 genes-12-00588-f002:**
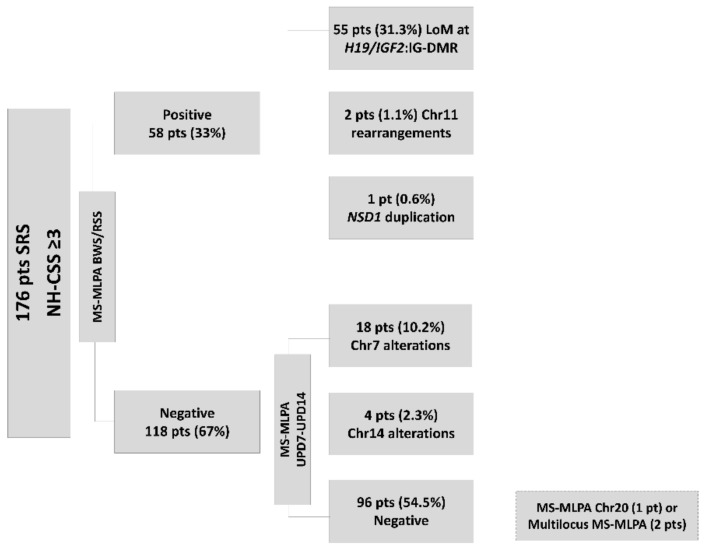
Flowchart for Silver Russell Syndrome (SRS)/SRS-like molecular diagnostics in our laboratory: 176 patients with Netchine–Harbison clinical scoring system (NH-CSS) ≥ 3 were first tested for methylation defects in the 11p15 region, resulting in 55 with LoM at *H19/IFG2*:IG-DMR, 2 with chromosome 11 rearrangements, and 1 with *NSD1* gene duplication. In the second step, methylation-specific multiplex ligation probe-dependent amplification (MS-MLPA) for chromosomes 7 and 14 revealed 17 cases of UPD(7)mat and 4 cases with Chr14 alterations (2 UPD(14)mat and 2 cases with LoM at *MEG3*-DMR). Negative patients were investigated for chromosome 20 uniparental disomy (UPD) using the approaches indicated in the rectangle framed by the broken lines.

**Figure 3 genes-12-00588-f003:**
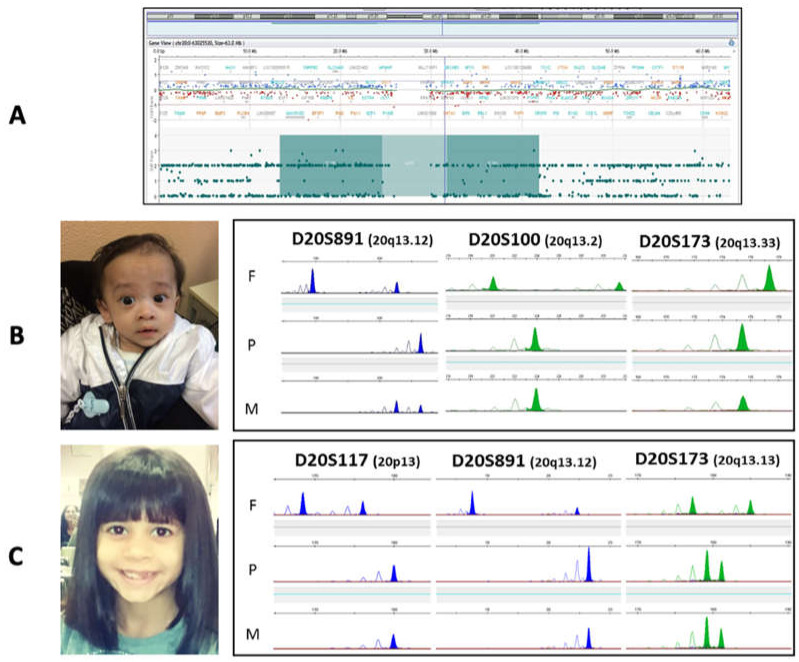
**Panel** (**A**): CGH+SNP array results of patient 1: the snapshot shows a balanced profile of the entire chromosome 20 and a 21Mb pericentromeric region of absence of heterozygosity. The AOH region encompasses 343 homozygous SNP polymorphisms. Of these, a total of 286 SNPs was informative on both the proband and the parents showed evidence of maternal isodisomy (dark green areas of the rectangle). **Panels** (**B**,**C**) show pictures of the patients with UPD(20) (on the left) and the STR analyses (on the right) of probands 2 and 3, respectively. The photos show patient 2 at the age of 5 months and patient 3 at the age of 8. Clinical features become more subtle after the first year of life. First infancy photos are not available.

**Table 1 genes-12-00588-t001:** Clinical features of the three new UPD(20)mat patients described here and previously reported cases.

Clinical Characteristics	Patient 1	Patient 2	Patient 3	Previously Reported Cases (*n* = 21)
Sex	M	M	F	11M/9F/1NA
Maternal age at birth (years)	44	42	40	38 (28–43)
Gestational age (weeks + days)	36 + 4	38 + 5	38 + 1	37 (32–40)
Oligohydramnios	NA	None	None	6/18
Birth length (cm | SDS)	43 | −2.27	44 | −2.65	46 | −1.7	–
Birth weight (kg | SDS)	1.51 | −3.12	2.23 | −2.38	2.48 | −1.79	–
Birth head circumference (cm | SDS)	30 | −2.53	33.1 | −1.01	31.5 | −1.96	–
Age at clinical SRS suspicion (months)	15	5.5	15	–
Evaluation length (cm | SDS)	71.5 | −2.7	60 | −2.7	69.9 | –2.5	–
Evaluation weight (kg | SDS)	6.7 | –4.9	4.15 | −5	6.52 | −4.9	–
UPD type	NA	UPhD	UPhD + UPiD	5/21 UPiD; 4/21 UPhD; 10/21 mixed
Methods	MS-MLPA + SNP-array	MS-MLPA + STRs	MS-MLPA + STRs	Various *
**NH-CSS ^1^ features**:				
SGA (birth weight and/or length ≤ –2 SDS)	+	+	–	15/21
Relative macrocephaly at birth ^a^	–	+	–	4/10
Postnatal growth failure ^b^	+ ^#^	+/–^$^	+ ^#^	18/21 ^#^
Protruding forehead ^c^	+	+	+	4/8
Body asymmetry ^d^	+	–	–	2/8
Feeding difficulties ^e^ and/or BMI ≤ –2 SDS (2ys)	+	+	+	18/21
**Other clinical manifestations**:				
Small and triangular face	+	+	+	6/8
Micrognathia	+	+	–	1/1
Hypotonia	+	–	+	9/13
Developmental delay	–	–	–	6/16
GH deficit	+	–	+	2/21
Facial dysmorphism	Blue sclera, ear anomalies	Epicanthus, helix hypoplasia, short philtrum, thin lips	Short palpebral fissures	11/21
Skeletal abnormalities	None	Fifth finger clinodactyly	None	13/19 (5/6 clinodactyly)
Genital anomalies	Cryptorchidism	None	Vaginal synechiae	3/3

^a^ Head circumference ≥ 1.5 SDS above birth weight and/or length SDS. ^b^ Height at 2 years ≤ –2 SDS or height ≤ –2 SDS below mid-parental target height. ^#^ The measurements were not taken at age 24 ± 1 months. ^$^ Borderline value. ^c^ Forehead projecting beyond the facial plane on a side view as a toddler (1–3 years). ^d^ Leg length discrepancy of ≥ 0.5 cm with at least two other asymmetrical body parts (one non-face). ^e^ Current use of a feeding tube or cyproheptadine for appetite stimulation. * 8/21: SNP array; 5/21: pyrosequencing + STR/SNP array; 1/21: MS-MLPA + SNP array/STR; 1/21: MS-PCR + STR; 1/21: diagnostic exome sequencing + STR; 4/21: STR; 1/21: allele-specific methylated multiplex real-time quantitative PCR. Abbreviations: NA = not available; UPhD = uniparental heterodisomy; UPiD = uniparental isodisomy.

## Data Availability

Not applicable.

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
