# Peer review of "Maternal Uniparental Disomy of Chromosome 20 (UPD(20)mat) as Differential Diagnosis of Silver Russell Syndrome: Identification of Three New Cases"

_genes, 2021, doi:10.3390/genes12040588_

Round 1
Reviewer 1 Report
Tannorella and cols describe three new cases of SRS-like caused by UPD(20)mat and compare their clinical caracteristics with those described in the 21 previous patients. This is an extra rare disorders so it is important to join efforts to define it properly.
There are just some minor aspects I would like to highlight:
Introduction:
- within minor contribution genetic defects, alterations at HGMA2 and PLAG1 genes should also be included
- I suggest authors to rewrite the sentence "UPD(20)mat causes the loss of the paternal GNAS imprinted locus at 20q13.22". It seems that GNAS locus is paternally imprinted, rather than the paternal imprinting is not present
Material and methods:
- patient 1: Instead of "For patient 1, array-CGH was performed using..." I suggest to indicate "For patient 1, an array was performed using..." as the array is not only for CGH but also for SNPs
Results:
- In the b/w printed versión of the mansucript information at figure 2 is very difficult to read
- In those CVNs including a DMR I suggest to mention the parental origin of the duplications
Author Response
Tannorella and cols describe three new cases of SRS-like caused by UPD(20)mat and compare their clinical characteristics with those described in the 21 previous patients. This is an extra rare disorders so it is important to join efforts to define it properly. There are just some minor aspects I would like to highlight
We thank the reviewer for appreciating the topic of our paper ; we followed his/her suggestions
Introduction: within minor contribution genetic defects, alterations at HGMA2 and PLAG1 genes should also be included
We mentioned the HGMA2/PLAG1/IGF2 pathway and inserted the relative reference
I suggest authors to rewrite the sentence "UPD(20)mat causes the loss of the paternal GNAS imprinted locus at 20q13.22". It seems that GNAS locus is paternally imprinted, rather than the paternal imprinting is not present
Our sentence was misleading and we modified it as suggested by the reviewer
Material and methods: patient 1: Instead of "For patient 1, array-CGH was performed using..." I suggest to indicate "For patient 1, an array was performed using..." as the array is not only for CGH but also for SNPs done, thank you
Results: In the b/w printed versión of the manuscript information at figure 2 is very difficult to read: The background and characters of the figure have been modified to enhance its readability
In those CVNs including a DMR I suggest to mention the parental origin of the duplications: done.
Reviewer 2 Report
The authors identified in their cohort of 176 patients fullifilling the critieria for Silver-Russell-syndrome diagnostics (Netchine-Harbinson-Score >= 3) 3 patients with UPD(20)mat syndrome (MULCHANDANI-BHOJ-CONLIN SYNDROME; MBCS, OMIM #617352). They conclude that their patients support the hypothesis that there is a clear overlap between SRS and MBCS and that for this reason in the diagnostic workup of patients who remain unremarkable in SRS screening, MBCS screening should be performed as consented by the SRS guidelines (Wakeling et al., 2017).
Although the authors do not present really new findings, the paper is of scientific interest because so far the clinical spectrum of MBCS has not been fully captured and the authors raise awareness to consider the clinical overlap of the different imprinting disorders in the diagnostic work-up. Overall, a streamlined presentation of the results would be desirable here.
My suggestions for improving the paper are:
- The title is a bit misleading. The authors should find a clear title that makes MBCS (UPD(20)mat) appear as a differential diagnosis rather than a "subtype" of SRS.
- The abstract should include the omim-annotated name of the MBCS and the omim-#number. For clinicians not fully familiar with imprinting syndromes, an introductory sentence would be helpful as an overview of the imprinting syndromes described until to date.
- Keywords should also include MBCS
- Introduction: It could be stronger emphasized that the flow chart of the SRS diagnostics already includes the differential diagnoses Temple syndrome, UPD(16)mat and UPD(20)mat according to the current consensus guidelines.
- The results of the study could be summarized in a separate subsection. The three figures could be merged into a single figure with a flow chart on the left and results (SNP-array-results, photos of two patients) on the right.
- For patient 1, open questions arise. GH deficiency is rather unusual for SRS and MBCS. Furthermore, the patient's sister is also GH-deficient! Should an additional genetic cause be considered at this point and further diagnostics be performed in both children? In UPDs the occurrence of body asymmetry is rather rare than in epimutations? Can the asymmetry actually be diagnosed with certainty in this patient according to the NHS Score LLD of ≥ 0.5 cm or arm asymmetry or LLD < 0.5 cm with at least two other asymmetrical body parts one non-face.
- Has patient 2 an Asian background? Line 249 weeks of gestation
- Conclusion: The first sentence of the Conclusion is not correct. The authors cannot ultimately say what the fourth most common pathomechanism is in their cohort when they studied "only" four pathomechanisms at all. To really say how common which mechanism is, a CNV analysis should have been done in the whole cohort and at the current state of knowledge a whole exome analysis on differential diagnoses. The first sentence must be omitted.

Author Response
My suggestions for improving the paper are:
- The title is a bit misleading. The authors should find a clear title that makes MBCS (UPD(20)mat) appear as a differential diagnosis rather than a "subtype" of SRS.
We thank the reviewer for the suggestion. We agree that Mulchandani-Bhoj-Conlin Syndrome is a differential diagnosis of Silver Russell, so we specified it in the title.
The abstract should include the OMIM-annotated name of the MBCS and the OMIM-#number: done. Inclusion of this correct specification made the abstract word number slightly exceeding: we hope the few extra words are tolerated
For clinicians not fully familiar with imprinting syndromes, an introductory sentence would be helpful as an overview of the imprinting syndromes described until to date:
Due to the requested limited number of words we could not add the suggested overview sentence of the imprinting syndromes in the abstract; however, the imprinting concept is explained in the introduction
- Keywords should also include MBCS: done, thank you
Introduction: It could be stronger emphasized that the flow chart of the SRS diagnostics already includes the differential diagnoses Temple syndrome, UPD(16)mat and UPD(20)mat according to the current consensus guidelines.
As suggested by the reviewer, we underlined the flow-chart mentioned in the consensus.
The results of the study could be summarized in a separate subsection. The three figures could be merged into a single figure with a flow chart on the left and results (SNP-array-results, photos of two patients) on the right.
We thank the reviewer for this constructive suggestion: we compacted within a single figure SNP-array results and patients’ photo, leaving apart the flow-chart which otherwise would become not readable
For patient 1, open questions arise. GH deficiency is rather unusual for SRS and MBCS. Furthermore, the patient's sister is also GH-deficient! Should an additional genetic cause be considered at this point and further diagnostics be performed in both children?
The referee is right, but the proband’s sister has never been evaluated by the clinician co-Authors of this study, because her reference pediatrician thought she did not need the attention and surveillance by medical geneticists. So we infer she is not dysmorphic. While we cannot absolutely exclude that the GH deficit in these sibs may arise from a different genetic cause, we notice that also in case 3 the GH values were borderline before the treatment.
In UPDs the occurrence of body asymmetry is rather rare than in epimutations? Yes, the occurrence of body asymmetry is rarer than in epimutations, but it is observed in about 30% of UPD7mat, as reported in the Supplementary of the SRS consensus. The asymmetry of patient 1 was “clinically apparent, from the end of the bed” as reported in (Kalish et al Am J Med Genet. 2017;173A:1735–1738)
Has patient 2 an Asian background? Line 249 weeks of gestation
We thank the reviewer for this comment: the boy originates from Philippines. We checked his growth parameters on Philippines growth charts (https://www.nnc.gov.ph/downloads/category/34-who-cgs-reference-table-0-71-mos) and the growth deficit is estimated severe according to these charts too.
Conclusion: The first sentence of the Conclusion is not correct. The authors cannot ultimately say what the fourth most common pathomechanism is in their cohort when they studied "only" four pathomechanisms at all. To really say how common which mechanism is, a CNV analysis should have been done in the whole cohort and at the current state of knowledge a whole-exome analysis on differential diagnoses. The first sentence must be omitted.
As requested we changed our sentence canceling the number fourth.